# The association between hyperlipidemia, lipid-lowering drugs and diabetic peripheral neuropathy in patients with type 2 diabetes mellitus

Kuo-Cheng Chang[1]ʘ, Yen-Wei Pai[1,2]ʘ, Ching-Heng Lin[3], I-Te Lee[4,5,6], Ming-Hong Chang [1,2]*

1 Neurological Institute, Taichung Veterans General Hospital, Taichung, Taiwan, 2 Department of Post-Baccalaureate Medicine, College of Medicine, National Chung Hsing University, Taichung, Taiwan, 3 Department of Medical Research, Taichung Veterans General Hospital, Taichung, Taiwan, 4 Division of Endocrinology and Metabolism, Department of Internal Medicine, Taichung Veterans General Hospital, Taichung, Taiwan, 5 Department of Medicine, School of Medicine, National Yang-Ming University, Taipei, Taiwan, 6 Department of Medicine, School of Medicine, Chung Shan Medical University, Taichung, Taiwan

ʘ These authors contributed equally to this work.
* cmh500809@gmail.com

**Data Availability Statement:** All relevant data are within the manuscript and its Supporting Information files.

## Abstract

### Aims

Previous studies showed conflicting relationship between hyperlipidemia, lipid-lowering therapy and diabetic peripheral neuropathy (DPN). As most of these works emerges from the Western and Australian countries, our study aims to investigate whether hyperlipidemia or lipid-lowering therapy (LLT) is associated with DPN in Taiwanese patients with type 2 diabetes (T2D).

### Methods

A cross-sectional, hospital-based observation study in adults with T2D was conducted from January to October 2013. DPN was screened using the Michigan Neuropathy Screening Instrument. Data were obtained at the time of enrollment, including medication usage, anthropometric measurements and laboratory examinations.

### Results

2,448 participants were enrolled, 524 (21.4%) of whom had DPN. Patients with DPN had significantly lower plasma total cholesterol (185.6 ± 38.6 vs 193.4 ± 42.3 mg/dL) and low-density lipoprotein cholesterol levels (114.6 ± 32.7 vs 119 ± 30.8 mg/dL). Multivariate analysis demonstrated that neither hyperlipidemia (adjusted OR (aOR), 0.81; 95% confidence interval (CI), 0.49–1.34) nor LLT (aOR, 1.10; 95% CI, 0.58–2.09) was associated with DPN. Subgroup analysis revealed that neither total cholesterol (aOR, 0.72; 95% CI, 0.2–2.62), low-density lipoprotein cholesterol levels (aOR, 0.75; 95% CI, 0.2–2.79), statin (aOR, 1.09; 95% CI, 0.59–2.03) nor fibrate (aOR, 1.73; 95% CI, 0.33–1.61) was associated with DPN.

**Funding:** This research received grants from Taichung Veterans General Hospital (TCVGH 1093404C and 1103402C), but nothing from commercial or not-for-profit sectors. The funders had no role in study design, data collection and analysis, decision to publish, or preparation of the manuscript.

**Competing interests:** The authors have declared that no competing interests exist.

## Conclusion

Our results suggest that neither hyperlipidemia nor lipid-lowering medication was associated with DPN in adults with T2D. DPN is a multifactorial disease, and our findings indicate that lipid metabolism may play a minor role in its pathogenesis.

## Introduction

Diabetic peripheral neuropathy (DPN) is the prevailing chronic complication in patients with type 1 (T1D) and type 2 diabetes (T2D). The lifetime prevalence of DPN can reach as high as 50% in adults with diabetes [1], and it contributes to numerous disabling morbidities, such as diabetic foot ulceration and consequent lower limb amputation [2]. The most common type of DPN is distal symmetric polyneuropathy, which accounts for approximately 75% of all DPN cases [1].

Because there is no known disease-modifying therapy for DPN other than serum glucose control, risk factor reduction in vulnerable populations is essential to limit future cases of DPN. The incidence of DPN in patients with T1D has the potential to be reduced by up to 60–70% with appropriate glycemic control measures [3, 4]; in contrast, preventing DPN by treating hyperglycemia in T2D yields only a 5–7% reduction in DPN incidence [5, 6]. Despite strict glucose control, more than 40% of diabetic patients are eventually diagnosed with peripheral neuropathy, suggesting that hyperglycemia is not the sole instigator of DPN [6]. The risk factors reported in previous studies include hyperglycemia, elevated glycated hemoglobin levels (HbA1c), increased duration of diabetes, insulin resistance, metabolic syndrome, hypertension, smoking, alcohol abuse, and abdominal obesity [1, 7]. Dyslipidemia may be another risk factor for DPN. Studies have found that high plasma triglycerides (TG) and low high-density lipoprotein cholesterol (HDL-C) levels were associated with DPN in patients with diabetes [8–10]. Unlike the well-established correlation between high TG and low HDL-C levels, the correlation is variable between DPN and other serum lipid components, such as total cholesterol (TC) and low-density lipoprotein cholesterol (LDL-C) levels. Some studies have found a positive correlation between elevated serum TC levels and DPN [11, 12], while others report that low LDL-C and TC levels are also risk factors for DPN [13, 14]. Furthermore, conflicting data have been reported on the impacts of dyslipidemia treatment on DPN. Two Danish studies found that treatment with either statin or fibrate was associated with a lower incidence of DPN [15, 16]. However, one nationwide study in the United States demonstrated that the prevalence of peripheral neuropathy was significantly higher among statin-users compared with non-users [17]. At the present time, there is insufficient evidence to advocate the association between lipid-lowering drugs and DPN in Asian population. The objective of the present study is to explore whether lipid-lowering therapy (LLT) or hyperlipidemia is associated with DPN among Han Taiwanese adults with T2D.

## Methods

### Study design and participants

This was a retrospective, cross-sectional observational study. Patients with prevalent or newly diagnosed T2D over 20 years of age between January 2013 and October 2013 were eligible for inclusion. Each of the participants was diagnosed by endocrinologists in the outpatient units at a Taiwanese tertiary medical center, which serves approximately 6,500 outpatients and 1,500 inpatients every day. All eligible patients were enrolled in our diabetes care network, hence

detailed demographic and clinical recordings were regularly documented. Before enrolment, the physicians and trained care-management nurse had conducted chart review and excluded those with history of chemotherapy exposure, alcohol abuse, or hereditary neuropathy. All enrolled participants were free from these conditions. The diagnoses of T2D were based on the criteria of American Diabetes Association (ADA). Individuals who were diagnosed with T1D or gestational diabetes, and whose plasma lipid measurements were not available were excluded. Before drawn for analysis, patients' information was anonymized by computer system, and the researchers were blinded to these data. The study was approved by the Institutional Review Board of Taichung Veterans General Hospital (CG18082B-1). Because of the retrospective and anonymous nature of this study, the need for informed consent was waived by Taichung Veterans General Hospital Institutional Review Board. All methods were performed in accordance with the relevant guidelines and regulations.

## Anthropometric measurements

Upon entry into this study, a case-management nurse performed anthropometric measurements to all participants. The anthropometric measurements included height, weight, waist circumference, and body mass index (BMI). Resting systolic and diastolic blood pressure were measured in the right arm when the patient was in a sitting position. For the details on the anthropometric measurements, please refers to our published studies [18–22]. Besides, the care-management nurse was also responsible for recording all of the eligible patients' previous or current disease status, duration of diabetes, hypoglycemic medication use, plasma lipid profiles, prescription of LLT (including statins and fibrates) and lifestyle habits, including smoking or alcohol drinking.

## Biochemical data

Laboratory examinations were administrated during endocrinology outpatient visits. In the morning after an overnight fasting period, blood samples were obtained from the antecubital vein. Fasting plasma glucose (using standard enzymatic methods), HbA1c (using high-performance liquid chromatography), and plasma lipid profiles (using standard enzymatic methods), including TC, HDL-C, LDL-C, and TG, were measured and recorded. To identify the individuals with hyperlipidemia, we adopted the following cut-off points: TG: > 200 mg/dL, TC: > 200 mg/dL, and LDL-C: > 130 mg/dL, according to the National Cholesterol Education Program Adult Treatment Panel III [23]. Using the six-variable Modification of Diet in Renal Disease equation [24], estimated glomerular filtration rate (eGFR) was estimated as:

$$186 \times \text{Serum creatinine}^{-1.154} \times \text{Age}^{-0.203} \times (0.742 \text{ if female})$$

For the patients who had received LLT, their baseline lipid levels were defined as the mean plasma lipid values (including TC, HDL-C, LDL-C and TG) in the 5 years prior to drug prescription. For the individuals who had never received LLT, the baseline lipid profiles were circumscribed as the 5-year mean plasma lipid levels prior to study enrollment, from 2008 to 2012. Meanwhile, the follow-up lipid levels were measured in both LLT users and non-users. In the LLT user group, the follow-up lipid values were defined as the mean lipid values in the 5 years after medication prescription. In the non-user group, the mean lipid values within the 5 years after study entry, from 2013 to 2017, were regarded as their follow-up lipid values.

## Diagnosis of hyperlipidemia and comorbidities

Diagnosis of hyperlipidemia depended on their plasma lipid profiles recorded when they were enrolled in our study. Individuals who had either hypercholesterolemia (TC > 200 mg/dL),

hyper-LDL-C (LDL-C > 130 mg/dL), or hypertriglyceridemia (TG > 200 mg/dL) were divided into the hyperlipidemic group. If the participants did not have hyperlipidemia, they were regarded as normolipidemic. Patients were categorized into three groups on the basis of their lipidemic profiles and usage of LLT. Those patients with lipid profiles (TG, LDL-C, and TC) that do not exceed the aforementioned testing thresholds and had never received LLT are considered normolipidemic non-LLT users. Those who administrated LLT were divided further depending on the diagnosis of hyperlipidemia into two groups, normolipidemic LLT users as well as hyperlipidemic LLT users.

Any of the following outpatient or inpatient claims recorded from 2010 to 2012 were considered as comorbidities. These comorbidities, which were also identified using the International Classification of Diseases, Ninth Revision, Clinical Modification (ICD-9-CM) codes, included cardiovascular disease (CVD; ICD-9-CM 390–438), ischemic heart disease (ICD-9-CM 410–414), and liver disease (ICD-9-CM 571–573). Both ischemic heart disease and cardiovascular disease were classified as co-morbid heart disease in our study.

## Assessment of diabetic peripheral neuropathy

All of the enrolled patients received assessment of DPN by the same trained and certificated care-management nurse to minimize the inter-rater reliability. DPN was evaluated based on the second component of Michigan Neuropathy Screening Instrument (MNSI). Based on the latest position statement of DPN by the American Diabetes Association (ADA), the MNSI was recommended as a validated instrument to identify the neuropathy end points for researches and clinical trials [25]. The MNSI was widely used in abundant clinical studies of patients with T2D [5, 8, 15, 26], including our published works [18–22]. Physical appearance of feet, ulceration, ankle deep tendon reflexes, and the perception of light touch (using Semmes-Weinstein 5.07 10-g monofilament) and distal vibration (using 128-Hz tuning fork) were investigated. As previous validated studies in adults [27], individuals whose MNSI examination (MNSIE) score more than two were assigned to the DPN group. Based on the current diagnostic criteria for DPN, those patients in the DPN group were supposed to be categorized as possible or probable DPN [28].

## Statistical methods

Descriptive statistics were presented as the mean values ± standard deviation (SD) and as the numbers with percentages. We used Fisher's exact test or chi-squared test to analyze categorical variables, while the analyses of continuous variables were conducted using ANOVA tests.

Multivariate logistic regression analyses were carried out to explore the effect of each identified independent variable on DPN. The multivariate regression models included all the confounders and the variables that had shown a significant correlation, and the adjusted odds ratios (OR) with 95% confidence interval (CI) were calculated between the comparison groups. The statistical significance level chosen was P value less than 0.05 (P < 0.05), and all tests were two-sided. All the data were analyzed using statistical package SAS version 9.4 for Windows.

Results

A total of 2,838 patients with diabetes were screened. After excluding 85 patients with type 1 diabetes or gestational diabetes, as well as 305 patients with missing serum lipid measurements, a final cohort of 2,448 patients was included in the analysis. A flow chart outlining patient selection is presented in Fig 1.

## Clinical and demographic characteristics

The sample's clinical and demographic characteristics are summarized in Table 1. A total of 2,448 patients were included in our study, 524 (21.4%) of whom were in the DPN group. The

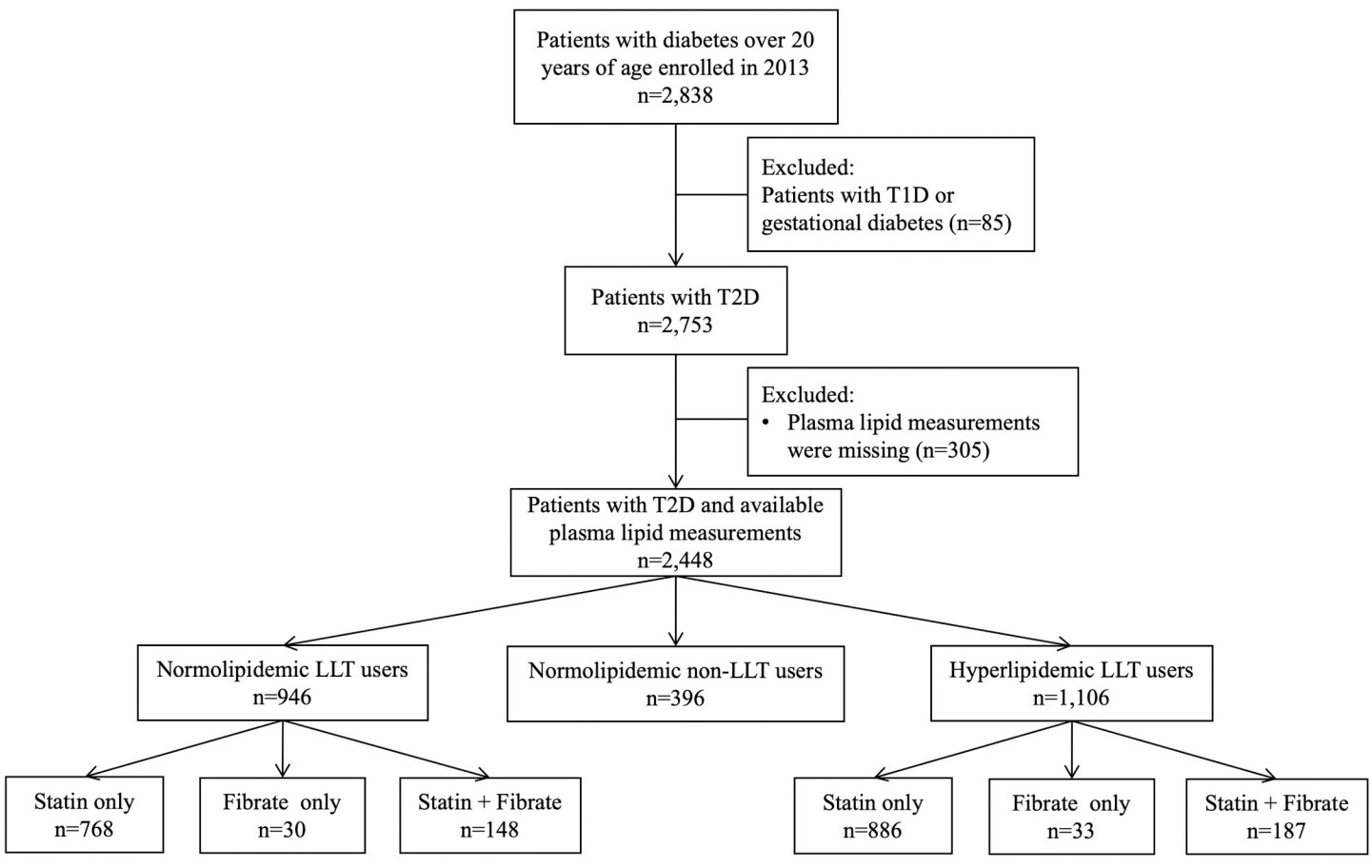

**Fig 1. Flow chart for patient selection in our study.**

mean age of participants was 64.0 ± 12.9 years at the time of the investigation. 1409 (57.6%) of the sample were male and 1039 (42.4%) were female. The mean duration of diabetes was 10.3 ± 8.4 years, and the mean serum HbA1c level was 7.4 ± 1.4% (57 ± 15.3 mmol/mol). The majority of patients (96.2%) had received at least one form of pharmacological treatment for diabetes.

The patients with DPN were significantly older than the patients without DPN (71.1 ± 12.4 years vs 62.1 ± 12.4 years). Compared to those without DPN, patients with DPN tended to have a larger waist circumference (92.6 ± 10.4 cm vs 90.4 ± 10.4 cm), were more likely to smoke (10.4% vs 6.7%), and had a longer duration of diabetes (13.9 ± 9.4 years vs 9.3 ± 7.9 years), fewer prescriptions for oral hypoglycemic medication, more insulin prescriptions, lower eGFR levels (65.1 ± 31.6 vs 81.5 ± 29.8 mL/min/1.73m$^2$), and higher rates of co-morbid heart disease (35.3% vs 15.4%). No differences were found between DPN patients and controls with respect to gender, BMI, history of alcohol consumption, systolic and diastolic blood pressure, fasting plasma glucose, HbA1c levels, and co-morbid liver disease. The proportion of patients using LLT was not significantly different between the two groups.

In respect to the plasma lipid levels, the patients with DPN had not only lower baseline and follow-up TC values (baseline: 185.6 ± 38.6 vs 193.4 ± 42.3 mg/dL; follow-up: 155.7 ± 28.2 vs 162.4 ± 28.1 mg/dL), but also lower baseline and follow-up LDL-C levels (baseline: 114.6 ± 32.7 vs 119 ± 30.8 mg/dL; follow-up: 89.7 ± 23.1 vs 94.8 ± 23.7 mg/dL). No differences were found in plasma HDL-C or TG levels.

**Table 1. Clinical and sociodemographic characteristics of the study.**

| Variable | Total (n = 2448) | Without neuropathy (n = 1,924) | With neuropathy (n = 524) |
|---|---|---|---|
| | N (%) | N (%) | N (%) |
| **Sociodemographic factors** | | | |
| Age, years | 64.0±12.9 | 62.1±12.4 | 71.1±12.4* |
| ≧65 | 1149 (46.9%) | 777 (40.4%) | 372 (71.0%)* |
| Male Gender | 1409 (57.6%) | 1090 (56.7%) | 319 (60.9%) |
| BMI, kg/m$^2$ | 25.7 ± 4.1 | 25.8 ± 4.1 | 25.4 ± 4.1 |
| Waist circumference, cm | 90.9 ± 10.5 | 90.4 ± 10.4 | 92.6 ± 10.4* |
| SBP≧130 mmHg or DBP≧85 mmHg | 1389 (57.4%) | 1075 (56.5%) | 314 (39.4%) |
| Smoker | 236 (9.6%) | 201 (10.4%) | 35 (6.7%)* |
| Alcohol drinker | 86 (3.5%) | 66 (3.4%) | 20 (3.8%) |
| **Diabetes-related factors** | | | |
| Fasting plasma glucose, mg/dL | 143.2 ± 46.3 | 143.4 ± 45.9 | 142.6 ± 48 |
| HbA1c, % (mmol/mol) | 7.4 ± 1.4 | 7.4 ± 1.4 | 7.5 ± 1.5 |
| Duration of diabetes, years | 10.3 ± 8.4 | 9.3 ± 7.9 | 13.9 ± 9.4* |
| Type of diabetes treatment | | | |
| No medication | 77 (3.8%) | 59 (3.7%) | 18 (4.0%)* |
| OHA only | 1430 (69.8%) | 1163 (72.6%) | 267 (59.9%)* |
| Insulin only | 156 (7.6%) | 100 (6.2%) | 56 (12.6%)* |
| Insulin + OHA | 385 (18.8%) | 280 (17.5%) | 105 (23.5%)* |
| Lipid-lowering drugs | | | |
| No medication | 396 (16.2%) | 298 (15.5%) | 98 (18.7%) |
| Statin only | 1654 (67.6%) | 1305 (67.8%) | 349 (66.6%) |
| Fibrate only | 63 (2.6%) | 47 (2.4%) | 16 (3.1%) |
| Statin + fibrate | 335 (13.7%) | 274 (14.2%) | 61 (11.6%) |
| **Biochemical factors** | | | |
| Baseline TC, mg/dL | 191.7 ± 41.6 | 193.4 ± 42.3 | 185.6 ± 38.6* |
| ≧200 | 878 (36.3%) | 709 (37.3%) | 169 (32.5%)* |
| Baseline TG, mg/dL | 199.7 ± 228.6 | 198.6 ± 223.1 | 204.5 ± 252.5 |
| ≧200 | 207 (30.0%) | 168 (29.8%) | 39 (31.0%) |
| Baseline HDL-C, mg/dL | 50.3 ± 13.2 | 50.3±12.9 | 50.2±14.1 |
| Female< 50; male< 40 | 749 (31.5%) | 582 (31.2%) | 167 (32.7%) |
| Baseline LDL-C, mg/dL | 118.0 ± 31.3 | 119 ± 30.8 | 114.6 ± 32.7* |
| ≧130 | 524 (31.5%) | 423 (32.7%) | 101 (27.5%) |
| Follow-up TC, mg/dL | 161.0 ± 28.3 | 162.4 ± 28.1 | 155.7 ± 28.2* |
| ≧200 | 187 (7.8%) | 154 (8.2%) | 33 (6.5%) |
| Follow-up TG, mg/dL | 151.1 ± 118.6 | 153.2 ± 123.4 | 143.1 ± 97.6 |
| ≧200 | 240 (17.9%) | 189 (17.7%) | 51 (18.5%) |
| Follow-up HDL-C, mg/dL | 50.2 ± 12.6 | 50.4 ± 12.5 | 49.8 ± 13.2 |
| Female<50; male<40 | 742 (31.1%) | 587 (31.2%) | 155 (30.6%) |
| Follow-up LDL-C, mg/dL | 93.7 ± 23.7 | 94.8 ± 23.7 | 89.7 ± 23.1* |
| ≧130 | 150 (6.3%) | 125 (6.7%) | 25 (4.9%) |
| eGFR | 77.9 ± 30.9 | 81.5 ± 29.8 | 65.1 ± 31.6* |
| < 60 ml/min/1.73m$^2$ | 603 (27.8%) | 388 (22.8%) | 215 (45.7%)* |
| **Comorbidities** | | | |
| Heart disease (CVD or IHD) | 482 (19.7%) | 297 (15.4%) | 185 (35.3%)* |

(*Continued*)

**Table 1.** (Continued)

| Variable | Total (n = 2448) | Without neuropathy (n = 1,924) | With neuropathy (n = 524) |
|---|---|---|---|
| | N (%) | N (%) | N (%) |
| Liver disease | 48 (2.0%) | 36 (1.9%) | 12 (2.3%) |

*P < 0.05

Data are expressed as mean ± SD for continuous variables and frequency (%) for categorical variables. Differences in continuous variables by ANOVA; differences in categorical variables by Fisher's exact or chi-squared test.

BMI, body mass index; kg, kilograms; cm, centimeters; HbA1c, glycated hemoglobin; OHA, oral hypoglycemic agent; SBP, systolic blood pressure; DBP, diastolic blood pressure; TC, total cholesterol; TG, triglyceride; HDL-C, high-density lipoprotein cholesterol; LDL-C, low-density lipoprotein cholesterol; eGFR, estimated glomerular filtration rate; CVD, cardiovascular disease; IHD, ischemic heart disease.

## Comparison of the association of DPN between normolipidemic non-LLT users, normolipidemic LLT users, and hyperlipidemic LLT users

As mentioned, the 2,448 patients were allocated to one of three groups: normolipidemic non-LLT users, normolipidemic LLT users, or hyperlipidemic LLT users, depending on the lipidemic status and prescription of lipid-lowering drugs. The normolipidemic non-LLT user group comprised 396 patients (16.2%), 98 of whom (24.75%) had DPN; the normolipidemic LLT user group comprised 946 patients (38.6%), 213 of whom (22.52%) had DPN; and the hyperlipidemic LLT user group comprised 1106 individuals (45.2%), 213 of whom (19.26%) had DPN. To determine whether hyperlipidemia or lipid-lowering drugs were associated with DPN, multivariate logistic regression analysis was performed by adjusting for all the prominent covariates shown in Table 1, including age, waist circumference, tobacco use, duration of T2D, type of diabetes treatment, eGFR, and heart disease. Although no significant differences were shown, gender was also adjusted since it is commonly considered an important sociodemographic factor. Multivariate analysis revealed no differences in the odds of DPN between the normolipidemic non-LLT users and normolipidemic LLT users (OR: 1.10; 95%CI: 0.58–2.09; Table 2). These results indicate that in patients without hyperlipidemia, the use of LLT did not affect the odds of DPN. Furthermore, among the hyperlipidemic LLT users, the odds

**Table 2. Adjusted odd ratios (ORs) and 95% confidence intervals (CIs) for DPN, according to lipidemic status and lipid-lowering drug use condition.**

| Variables | Adjusted OR[a] | 95% CI |
|---|---|---|
| Sub-analysis | | |
| Normolipidemic non-LLT users | 1.00 | — |
| Normolipidemic LLT users | 1.10 | 0.58–2.09 |
| Hyperlipidemic LLT users | 0.89 | 0.45–1.76 |
| Sub-analysis | | |
| Normolipidemic LLT users | 1.00 | — |
| Normolipidemic non-LLT users | 0.91 | 0.48–1.72 |
| Hyperlipidemic LLT users | 0.81 | 0.49–1.34 |

LLT, lipid-lowering therapy

[a]Adjusted for age, gender, waist, smoking, duration of diabetes, type of diabetes treatment, eGFR, heart disease. The multivariate model included all variables that were significantly different between patients with or without DPN in Table 1. Type of diabetes treatment was classified and adjusted into four groups: lifestyle modification, oral hypoglycemic agent (OHA) only, insulin only, combination of OHA and insulin. As an important demographic factor, gender was also adjusted.

of DPN did not significantly differ from those in the normolipidemic non-LLT user or normolipidemic LLT user group, suggesting that neither hyperlipidemia nor lipid-lowering drugs significantly influenced the odds of DPN in patients with T2D.

## Differential impacts of LDL-C or TC on DPN in normolipidemic non-LLT users, normolipidemic LLT users, and hyperlipidemic LLT users

We planned to define whether there is any different impact of each lipid profile on DPN. As mentioned in Table 1, the baseline and follow-up plasma levels of LDL-C and TC were significantly lower in the DPN population in our study. Meanwhile, regarding plasma HDL-C and plasma TG, there was no difference between patients with and without DPN. Based on these findings, we decided to investigate the impacts of plasma LDL-C or TC, but not HDL-C or TG, on DPN. To explore whether plasma LDL-C or TC concentrations differentially affected the odds of DPN, patients with hyperlipidemic were subcategorized into two groups on the basis of plasma LDL-C and TC values. Patients with high plasma LDL-C levels (>130 mg/dL) were allocated to the hyper-LDL-C group, whereas those with low levels (<130 mg/dL) were assigned to the normal LDL-C group. Meanwhile, individuals with high plasma TC (>200 mg/dL) and normal plasma TC (<200 mg/dL) levels were assigned to the hyper-TC and normal TC group, respectively.

Multivariate logistic regression analysis was performed to determine whether baseline plasma LDL-C or TC levels were independently correlated with DPN. Relative to the normal TC non-LLT users, no significant differences were found with regard to DPN among either the normal TC LLT users or the hyper-TC LLT users (Table 3). The corresponding adjusted ORs of DPN were 1.09 (95% CI: 0.58–2.05) and 0.85 (95% CI: 0.42–1.74), respectively.

**Table 3. Adjusted odd ratios (ORs) and 95% confidence intervals (CIs) for DPN, according to classification of lipid profiles.**

| Variables | Adjusted OR[a] | 95% CI |
|---|---|---|
| **Sub-analysis–Baseline TC** | | |
| Baseline normal TC non-LLT users | 1.00 | — |
| Baseline normal TC LLT users | 1.09 | 0.58–2.05 |
| Baseline hyper-TC LLT users | 0.85 | 0.42–1.74 |
| **Sub-analysis–Baseline LDL-C** | | |
| Baseline normal LDL-C non-LLT users | 1.00 | — |
| Baseline normal LDL-C LLT users | 1.42 | 0.72–2.8 |
| Baseline hyper-LDL-C LLT users | 1.37 | 0.61–3.07 |
| **Sub-analysis–Follow-up TC** | | |
| Follow-up normal TC non-LLT users | 1.00 | — |
| Follow-up normal TC LLT users | 1.10 | 0.59–2.05 |
| Follow-up hyper-TC LLT users | 0.72 | 0.2–2.62 |
| **Sub-analysis–Follow-up LDLC** | | |
| Follow-up normal LDL-C non-LLT users | 1.00 | — |
| Follow-up normal LDL-C LLT users | 1.09 | 0.58–2.03 |
| Follow-up hyper-LDL-C LLT users | 0.75 | 0.2–2.79 |

TC, total cholesterol; LDL-C, low-density lipoprotein cholesterol; LLT, lipid-lowering therapy

[a]Adjusted for age, gender, waist, smoking, duration of diabetes, type of diabetes treatment, eGFR, heart disease. The multivariate model included all variables that were significantly different between patients with or without DPN in Table 1. Type of diabetes treatment was classified and adjusted into four groups: lifestyle modification, oral hypoglycemic agent (OHA) only, insulin only, combination of OHA and insulin. As an important demographic factor, gender was also adjusted.

Similarly, no differences in DPN odds were found between the normal LDL-C non-LLT users, the normal LDL-C LLT users (OR: 1.42, 95%CI: 0.72–2.8), and the hyper-LDL-C LLT users (OR: 1.37, 95% CI: 0.61–3.07).

Regarding follow-up plasma TC and LDL-C levels, similar results were found. Multivariate analysis revealed no differences in the odds of DPN between the normal TC non-LLT users, the normal TC LLT users (OR: 1.10, 95% CI: 0.59–2.05), and the hyper-TC LLT users (OR: 0.72, 95% CI: 0.2–2.62). Lastly, compared with the normal LDL-C non-LLT users, neither the normal LDL-C LLT users nor the hyper-LDL-C LLT users showed significant differences in the odds of DPN, with corresponding adjusted ORs of 1.09 (95% CI: 0.58–2.03) and 0.75 (95% CI: 0.2–2.79), respectively. These results suggest that plasma LDL-C and TC levels were not associated with DPN in our study.

### Differential impacts of statin or fibrate on DPN in the normolipidemic non-LLT users, the normolipidemic LLT users, and the hyperlipidemic LLT users

To determine whether different classes of LLT had diverse effects on the association of DPN, we categorized the patients who received lipid-lowering medication intervention into two sub-groups: the statin and the fibrate group. The results of the multivariate logistic regression analysis are shown in Table 4.

Compared to the normolipidemic non-LLT users, no significant differences in the odds of DPN were found in the normolipidemic statin users and the hyperlipidemic statin users. The corresponding adjusted ORs were 1.09 (95% CI: 0.59–2.03) and 0.89 (95% CI: 0.46–1.71) in the normolipidemic statin user and the hyperlipidemic statin user groups, respectively. As concerns fibrate use, no group was significantly associated with DPN. Our results suggest that neither statin nor fibrate are significantly associated with DPN.

### Discussion

To our knowledge, this is the first and largest cross-sectional study primarily aimed at investigating the association between hyperlipidemia and lipid-lowering drugs to the development of

**Table 4. Adjusted odd ratios (ORs) and 95% confidence intervals (CIs) for DPN, according to different classes of lipid-lowering drugs use.**

| Variables | Adjusted OR[a] | 95% CI |
|---|---|---|
| **Sub-analysis–Statin** | | |
| Normolipidemic non-LLT users | 1.00 | — |
| Normolipidemic statin users | 1.09 | 0.59–2.03 |
| Hyperlipidemic statin users | 0.89 | 0.46–1.71 |
| **Sub-analysis–Fibrate** | | |
| Normolipidemic non-LLT users | 1.00 | — |
| Normolipidemic fibrate users | 0.73 | 0.33–1.61 |
| Hyperlipidemic fibrate users | 0.50 | 0.16–1.6 |

LLT, lipid-lowering therapy

[a]Adjusted for age, gender, waist, smoking, duration of diabetes, type of diabetes treatment, eGFR, heart disease. The multivariate model included all variables that were significantly different between patients with or without DPN in Table 1. Type of diabetes treatment was classified and adjusted into four groups: lifestyle modification, oral hypoglycemic agent (OHA) only, insulin only, combination of OHA and insulin. As an important demographic factor, gender was also adjusted.

DPN in adults with T2D. Our study demonstrated that neither hyperlipidemia nor the use of lipid-lowering drugs was significantly associated with DPN in T2D patients. This finding remained consistent in patients with either hyper-TC, hyper-LDL-C, or who received statin or fibrate treatment. Moreover, patients with DPN in our study tended to have lower serum TC and LDL-C levels than those without DPN.

Compared with previous studies, our study consisted of more T2D patients who were under LLT [29]. 81.3 percent of our patients received statin alone or in combination with fibrate. There were 946 patients who were normolipidemic but still received LLT, mostly statin. The possible explanation was that statin was regarded as a cornerstone of prevention of atherosclerotic disease in patients with T2D. Based on the guideline of blood cholesterol management from American College of Cardiology/American Heart Association (ACC/AHA) in 2013, statin is recommended in diabetic patients whose serum LDL levels are between 70 to 189 mg/dL [30]. This recommendation is mainly aimed to reduce the atherosclerotic cardiovascular risk in patients with diabetes. Therefore, in our study, many patients might still receive statin even they did not have hyperlipidemia. However, the aforementioned subgroup analysis showed that there was no difference in the odds of DPN between the normolipidemic patients who received LLT or not. This finding implicated that statin was proven to have benefits in risk reduction in macrovascular complications, but when it comes to microvascular complications, this effect might be obscure.

Preclinical studies have described diverse pathophysiological mechanisms, resulting from increased intracellular glucose level, elevated serum free fatty acids, oxidized or glycated lipoproteins, in the development and progression of DPN [7, 31, 32]. Despite the molecular evidence implicating hyperlipidemia in the etiopathogenesis of DPN, clinical research yields conflicting results on the correlation between hyperlipidemia and DPN. The Utah Diabetic Neuropathy study, which enrolled 218 T2D patients who either had no symptoms of neuropathy or had symptoms for fewer than 5 years, showed that hypertriglyceridemia (defined as plasma TG $\geq$150mg/dL) was associated with an increased risk of DPN, especially small fiber neuropathy [33]. However, in this study, DPN was not defined by an identical diagnostic modality. Variable investigating techniques, including the Utah early neuropathy scale, nerve conduction studies, quantitative sensory tests, quantitative sudomotor axon reflex tests, or intraepidermal nerve fiber density (IENFD), were used to identify DPN. Using this criterion, the prevalence of DPN might be overestimated because the presence of neuropathic symptoms was not strictly required. Another study which enrolled 1,111 patients (767 with T2D, 344 with T1D) and DPN was identified by Neuropen [34]. Only those with painful DPN had hypertriglyceridemia, while plasma levels of TC, TG, and LDL-C were not associated with DPN in patients without neuropathic pain. In contrast, a study performed magnetic resonance neurography in 100 patients with T2D and found that low plasma TC and LDL-C levels significantly contributed to the severity of diabetic neuropathic lesions [13]. These findings aligned with some of ours, namely, that we also found that participants with DPN had significantly lower baseline and follow-up TC and LDL-C levels than those without DPN. A plausible explanation may be that lower cholesterol levels lead to an insufficient cholesterol supply for neurite regeneration following diabetic neuropathic injury. However, all of these aforementioned researches were from the United States or European countries and had a smaller population size, which might contribute to the inconsistent findings. A recent meta-analysis of 39 randomized controlled trials investigating the relationship between DPN and hyperlipidemia demonstrated that patients with DPN had higher TG and lower HDL-C levels relative to controls [35]. No significant differences in TC and LDL-C levels were found between the DPN and control group. Subgroup analysis clarified that only the patients with both T1D and DPN had higher TG levels than patients in the control group, whereas serum TG levels in T2D did

not differ from those of controls. This meta-analysis also illustrated that reduced serum TC and LDL-C levels may indicate a transition from an asymptomatic condition to a severe pain condition in patients with DPN. It is worth mentioning that, by providing a larger sample size, our study reported findings similar to those of the meta-analysis.

Meanwhile, the evidence in support of DPN management by lowering serum lipid levels remains insufficient and controversial [32]. Previous observational studies supported that lipid-lowering drugs were protective against DPN. In 2014, a Danish T2D cohort study of 15,679 prevalent statin users and 47,037 non-statin users found that prevalent statin users had a 35% risk reduction of developing DPN [16]. In 2019, a Taiwanese study investigating the effects of statin on diabetic triopathy in T2D patients found a 15% reduced risk of new-onset DPN among statin users relative to non-statin users [36]. However, a larger Danish T2D cohort study published in 2020 that enrolled 59,255 new statin users, 75,528 prevalent statin users, and 124,842 non-users found no evidence between statin therapy and risk of DPN. Moreover, they even showed a temporarily increased risk of DPN within the first year after statin initiation among new statin users (adjusted hazard ratio: 1.30, 95% CI: 1.12–1.53) [37]. Our research also supported the lack of meaningful correlation between hyperlipidemia and DPN after correcting for confounding factors (e.g., duration of diabetes). Furthermore, lipid-lowering drugs were neither positively nor negatively associated with DPN in our study. In the subgroup analysis focusing on the impacts of different lipid spectrum (including TC and LDL-C) and different types of LLT (including statin and fibrate) on DPN, our results remain consistent. Ultimately, our findings do not support the hyperlipidemia-related pathogenesis in DPN development.

One of the advantages of our research design is its large sample size. In fact, compared to other research efforts aimed at investigating the association between hyperlipidemia and DPN among patients with T2D, we provided a single-centered study with the largest population size. The other advantages of our study are the use of standardized and published data collection protocols and the inclusion of detailed information regarding confounding factors, laboratory data, and the treatment of T2D and hyperlipidemia. Furthermore, we adopted the MNSI, a validated clinical screening tool of DPN that is sensitive, specific, non-invasive, and easily administered by a single trained professional.

Our study also had several limitations. First, statin and fibrate were the only lipid-lowering drugs studied; new generation lipid-lowering drugs were not included, such as ezetimibe or proprotein convertase subtilisin/kexin type 9 (PCSK9) inhibitors. Second, the diagnosis of DPN was not corroborated by other laboratory techniques, such as nerve conduction studies, corneal confocal microscopy, laser Doppler flare imaging, sudomotor axon reflex assessment, quantitative sensory testing, and skin biopsy. Although the latest ADA position statement of DPN did not rigorously require electrophysiological examination for DPN screening, it is a requisite for diagnosis of confirmed DPN [25, 28]. Based on the current diagnostic criteria for DPN, an abnormal nerve conduction study and neuropathic symptoms and/or signs are essential to diagnose confirmed DPN. Clinical symptoms and signs without electrophysiological evidence only contribute to the diagnosis of possible or probable DPN [38]. Without electrophysiological corroboration, we instead used the MNSI to identify the patients who had possible or probable DPN. Besides, although some patients with DPN might demonstrated overt neurological symptoms, up to 50 percentage of DPN may be asymptomatic. In order to broadly assess the cases with incidental or subclinical DPN in a less-invasive procedure and to prevent further neuropathic injuries, we adopted the second portion of the MNSI for DPN investigation. Third, given the ethnic homogeneity of our sample, the results may not be representative of all diabetic patients. Fourth, as a cross-sectional study, it is inherently difficult to confirm causal relationship. Besides, reverse causality, which the presence of DPN may affect

lipid-lowering drug use or other factors under investigation, cannot be completely ruled out. Fifth, although efforts were made to adjust for potential confounding factors, there is a possibility of residual confounders influencing the observed associations. Despite these limitations, our study utilized standardized data collection procedures and detailed information on confounding factors, laboratory test values, and diabetes treatment to minimize residual confounding. We also used a validated clinical screening instrument (MNSI) to reduce misclassification of DPN.

## Conclusion

Our study demonstrated that neither hyperlipidemia nor lipid-lowering agent was associated with DPN in patients with T2D. Although preclinical studies suggest that hyperlipidemia may be involved in the pathogenesis of DPN, this hypothesis was not supported by our data. DPN is a complex and multifactorial microvasculopathy attributed to diverse mechanisms. Our findings suggest that hyperlipidemia may not be a significant independent risk factor for DPN in patients with T2D. Further longitudinal studies are warranted to test the causality of hyperlipidemia and DPN. Our study also found a trend of lower TC and LDL-C levels in patients with DPN. However, future research is needed to explore the potential independent association between low TC or LDL-C levels and the risk of DPN.

## Supporting information

**S1 Dataset.**
(XLS)

## Acknowledgments

The authors appreciate the volunteer's participation in this study, and the statistical support by Biostatics Taskforce of Taichung Veterans General Hospital.

## Author Contributions

**Conceptualization:** Yen-Wei Pai, I-Te Lee, Ming-Hong Chang.

**Data curation:** Yen-Wei Pai, I-Te Lee.

**Formal analysis:** Ching-Heng Lin.

**Funding acquisition:** Ming-Hong Chang.

**Investigation:** Yen-Wei Pai.

**Methodology:** Kuo-Cheng Chang, Ching-Heng Lin.

**Supervision:** I-Te Lee, Ming-Hong Chang.

**Validation:** Ming-Hong Chang.

**Writing – original draft:** Kuo-Cheng Chang.

**Writing – review & editing:** Yen-Wei Pai, Ming-Hong Chang.

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
