## [Decision Letter · Decision Letter 0]

1 Dec 2022

PONE-D-22-23997The association between hyperlipidemia, lipid-lowering drugs and diabetic peripheral neuropathy in patients with type 2 diabetes mellitusPLOS ONE

Dear Dr. Chang,

Thank you for submitting your manuscript to PLOS ONE. After careful consideration, we feel that it has merit but does not fully meet PLOS ONE’s publication criteria as it currently stands. Therefore, we invite you to submit a revised version of the manuscript that addresses the points raised during the review process.

We look forward to receiving your revised manuscript.

Kind regards,

Alok Raghav, PhD

Academic Editor

PLOS ONE

Journal Requirements

Additional Editor Comments:

Address the major concerns raised by the reviewer

Reviewers' comments:

Reviewer's Responses to Questions

**Comments to the Author**

1. Is the manuscript technically sound, and do the data support the conclusions?

Reviewer #1: Yes

Reviewer #2: Yes

2. Has the statistical analysis been performed appropriately and rigorously? 

Reviewer #1: Yes

Reviewer #2: Yes

3. Have the authors made all data underlying the findings in their manuscript fully available?

Reviewer #1: Yes

Reviewer #2: Yes

4. Is the manuscript presented in an intelligible fashion and written in standard English?

Reviewer #1: Yes

Reviewer #2: Yes

5. Review Comments to the Author

Reviewer #1: In the present study, the authors analyzed the association between hyperlipidemia, lipid-lowering drugs and diabetic peripheral neuropathy (DPN) . Their results showed that neither hyperlipidemia nor lipid-lowering medication was associated with DPN in adults with T2DM. The study population was large and the design was reasonable. But, I still have some concerns to address.

1.the authors analyzed the association of hyper-LDL and hyper-TC with risk of DPN. But why the association of hypertriglyceridemia and low-HDL with DPN were not evaluated. Although the two lipids were not different between two groups, the comparison might be confunded by other factors.

2.Since several studies have shown that hyperlipidemia, including LDL, TG, HDL, were contributors for DPN in T2DM (e.g. a review by Pop-Busui R, et al. Diabetes Care. 2017;40(1):136-54), the authors should discuss more details about the discrepancy (e.g. the differences in examination method, characteristics of subjects, statistical method... )

3.In the multi-regression analysis, some factors were adjusted (Table 2, 3,4). It should be explained why these factors were selected to adjust. Especially, how to adjust the type of diabetes treatment? I think the table and legends could be more detailed.

4.How was the sample size calculated?

5.How about the liver and renal functions (e.g.ALT, AST, Cr) in each group? Whether the patients with severe abnormal liver or renal functions were excluded, since the abnormal function could affect the blood lipid levels.

Reviewer #2: # Review

The manuscript is written clearly. A large sample size is included. A thorough statistical analysis is performed. The data are clearly presented and justify the conclusions. The references are up-to-date.

6. PLOS authors have the option to publish the peer review history of their article (what does this mean?). If published, this will include your full peer review and any attached files.

Reviewer #1: **Yes: **Qian Ge

Reviewer #2: No

---

## [Author Response · Author response to Decision Letter 0]

11 Jan 2023

Dear Editors and Reviewers: 

Thank you very much for your kind consideration of the publication of our article entitled “The association between hyperlipidemia, lipid-lowering drugs and diabetic peripheral neuropathy in patients with type 2 diabetes mellitus”. We deeply appreciate the editors’ and reviewers’ valuable comments and have revised our manuscript based on those opinions. For your convenience, the responses to the referees’ comments are shown in red color, and the changes are also shown in red color in the revised version of the manuscript. We would be glad to respond quickly to any further questions or comments.

Best regards,

Ming-Hong Chang, MD

Answers to Editor’s and Reviewers’ comments

Reviewer #1: In the present study, the authors analyzed the association between hyperlipidemia, lipid-lowering drugs and diabetic peripheral neuropathy (DPN). Their results showed that neither hyperlipidemia nor lipid-lowering medication was associated with DPN in adults with T2DM. The study population was large and the design was reasonable. But, I still have some concerns to address.

1. The authors analyzed the association of hyper-LDL and hyper-TC with risk of DPN. But why the association of hypertriglyceridemia and low-HDL with DPN were not evaluated. Although the two lipids were not different between two groups, the comparison might be confounded by other factors.

Ans. Thank you very much for your kind and useful suggestion to make the manuscript more comprehensive. In univariate analysis, we investigated the association of each lipid profile with the risk of DPN. As shown in Table 1, the baseline and follow-up plasma levels of LDL-C and TG were significantly lower in the DPN group in our study. Meanwhile, regarding plasma levels of HDL-C and TG, there was no difference between patients with and without DPN. Based on these findings, we hypothesized that plasma LDL-C or TC, but not HDL-C or TG, were associated with DPN. Further multivariate analysis was adjusted for all the variables that were significantly different between patients with or without DPN in Table 1. The revision is shown in red color at lines 234-240 in the revised manuscript. 

2.Since several studies have shown that hyperlipidemia, including LDL, TG, HDL, were contributors for DPN in T2DM (e.g. a review by Pop-Busui R, et al. Diabetes Care. 2017;40(1):136-54), the authors should discuss more details about the discrepancy (e.g. the differences in examination method, characteristics of subjects, statistical method... )

Ans. Thank you very much for your kind and useful suggestion to make the manuscript more comprehensive. We reviewed previous studies that investigated the association of hyperlipidemia and DPN. The comparison between each study is listed below. The limitations of these studies included small sample size, race disparity (mostly Caucasian population), and discrepancy in examination method. The strength of our study included large sample size, standardized data collection, and validated screening tool. The limitation of our study also included race disparity (mostly Han Chinese population). A recent meta-analysis (Zixin Cai et al.) provided similar results with our findings. It implies that with a larger sample size and Han Chinese population included, the association between hyperlipidemia and DPN is not significant. The revision is shown in red color at lines 321-327, 333-336 and 344-345 in the revised manuscript.

3. In the multi-regression analysis, some factors were adjusted (Table 2, 3, 4). It should be explained why these factors were selected to adjust. Especially, how to adjust the type of diabetes treatment? I think the table and legends could be more detailed.

Ans. Thank you very much for your kind and useful suggestion to make the manuscript more comprehensive. The table legends and results section are revised to include more details regarding the covariate adjustment. The revision is shown in red color at lines 211-217, 229-230, 261-262, and 282-283 in the revised manuscript. 

4. How was the sample size calculated?

Ans. Thank you very much for your precious comments. In this study, we enrolled the patients that fulfilled the inclusion criteria, which constituted our sample size. We believe that with this sample size, our results are convincing. 

5. How about the liver and renal functions (e.g. ALT, AST, Cr) in each group? Whether the patients with severe abnormal liver or renal functions were excluded, since the abnormal function could affect the blood lipid levels.

Ans. Thank you very much for your precious comments. In this study, liver function was assessed by liver disease, based on ICD-9-CM 571 to 573. Renal function was measured by estimated glomerular filtration rate (eGFR). The prevalence of liver disease showed no difference between patients without or with DPN, as shown in Table 1. Patients with DPN had significantly lower eGFR than those without DPN. We did not exclude those patients who had severe abnormal renal functions; instead, eGFR was adjusted in the multivariate regression analysis. 

Reviewer #2: # Review

The manuscript is written clearly. A large sample size is included. A thorough statistical analysis is performed. The data are clearly presented and justify the conclusions. The references are up-to-date.

Ans. Thank you so much for your kind comments. We are so grateful.

---

## [Decision Letter · Decision Letter 1]

3 Apr 2023

PONE-D-22-23997R1

The association between hyperlipidemia, lipid-lowering drugs and diabetic peripheral neuropathy in patients with type 2 diabetes mellitus

PLOS ONE

Dear Dr. Chang,

Thank you for submitting your manuscript to PLOS ONE. After careful consideration, we feel that it has merit but does not fully meet PLOS ONE’s publication criteria as it currently stands. Therefore, we invite you to submit a revised version of the manuscript that addresses the points raised during the review process.

As the original Academic Editor and one of the reviewers became unavailable, we sought input from additional reviewers. These reviewers provide different perspectives, which we now invite you to address and/or rebut, as we feel that in doing so the clarity and messaging of the manuscript will improve. Please see the full reports below.

We look forward to receiving your revised manuscript.

Kind regards,

Hanna Landenmark

Staff Editor

PLOS ONE

Reviewers' comments:

Reviewer's Responses to Questions

**Comments to the Author**

1. If the authors have adequately addressed your comments raised in a previous round of review and you feel that this manuscript is now acceptable for publication, you may indicate that here to bypass the “Comments to the Author” section, enter your conflict of interest statement in the “Confidential to Editor” section, and submit your "Accept" recommendation.

Reviewer #2: (No Response)

Reviewer #3: (No Response)

Reviewer #4: (No Response)

Reviewer #5: (No Response)

2. Is the manuscript technically sound, and do the data support the conclusions?

Reviewer #2: (No Response)

Reviewer #3: No

Reviewer #4: Yes

Reviewer #5: Yes

3. Has the statistical analysis been performed appropriately and rigorously? 

Reviewer #2: (No Response)

Reviewer #3: No

Reviewer #4: Yes

Reviewer #5: Yes

4. Have the authors made all data underlying the findings in their manuscript fully available?

Reviewer #2: (No Response)

Reviewer #3: No

Reviewer #4: No

Reviewer #5: Yes

5. Is the manuscript presented in an intelligible fashion and written in standard English?

Reviewer #2: (No Response)

Reviewer #3: Yes

Reviewer #4: No

Reviewer #5: Yes

6. Review Comments to the Author

Reviewer #2: The Authors of the manuscript entitled "The association between hyperlipidemia, lipid-lowering drugs and diabetic peripheral neuropathy in patients with type 2 diabetes mellitus" have made corrections according to the Reviewer's suggestions.

Reviewer #3: Dear Authors and Editor,

the main problem of this manuscript is the cross-sectional analyses. Such design does not allow to evalaute any causal link between lipid profile or lipid lowering therapy and DPN. Indeed this study is biased by several factors, e.g. confounding by indications, reverse causality, immortal time bias....

This design does not allow to address the role of lipid profile and LLT on DPN.

The authors should design a longitudinal study, evaluating onset of DPN after initiation of LLT or after exposure to low or high levels of HDL,TG,LDL etc...

The results therefore do not support the conclusions.

Reviewer #4: Thank you for giving me the opportunity to read your paper. This is an interesting cross-sectional study of the association between hyperlipidemia, lipid-lowering drugs and diabetic peripheral neuropathy (DPN) in patients with type 2 diabetes mellitus (T2D). Despite studies conducted among Western cohorts have shown that lipid-lowering medication is not associated with DPN, the evidence is insufficient for Asian cohorts. The study included patients with type 2 diabetes seen in an outpatient clinic at a Taiwanese tertiary medical center. Upon enrollment baseline characteristics were collected including anthropometric measures, medical history, and medication. Neuropathy was evaluated using measurements recommend by clinical guidelines including answering the Michigan Neuropathy Screening Instrument and a thorough clinical examination. Using regression analyses, the authors report that neither hyperlipidemia nor lipid-lowering drugs were associated with a risk of DPN. Despite these findings, the study has limitations – please see my comments below.

Major comments:

For those in lipid-lowering drug treatment, the authors obtained the baseline lipid level within five years before the drug was prescribed. I understand that patients in lipid-lowering therapy used the medication at baseline. What was the purpose by defining the baseline lipid level as the level before drug treatment and not using the on-treatment lipid level? The on-treatment lipid level may be more informative since lipid-lowering therapy per definition impacts the lipid level. Furthermore, is it possible to obtain information on how long time patients have received lipid-lowering therapy?

The cut-offs for triglycerides are very high. Other studies have used 150 mg/dL (1.7 mmol/L). 200 mg/dL is a target provided for clinicians who should consider initiating statin therapy in patients who have a high risk of CVD (Mach et al . 2019, 2019 ESC/EAS guidelines for the management of dyslipidaemias: Lipid modification to reduce cardiovascular risk). The cut-off is therefore not DPN specific. Furthermore, using cut-off points can be informative, but may also be problematic because the categorization implies an assumption about patients in a giving category have the same risk which may lead to loss of information (Royston et al. Stat med. 2006). While this assumption maybe approximately true in some cases, it is likely rare that this would hold biologically. Did the authors consider to use more flexible models to be inform the reader about the form of the association for example by using restricted cubic splines?

It is highly problematic that the authors categorize patients into “hyperlipidemic” subgroups based on three very different lipid measurements. While LDL cholesterol is important as a cardiovascular disease risk marker, triglycerides may be the most important for neuropathy. When categorized according to EITHER cholesterol OR triglycerides the results will tend to be diluted because LDL cholesterol is not associated with neuropathy. The authors should therefore consider regrouping the patients and analyze the results for each lipid parameter separately. At last, it is unclear why the authors repeat the analysis in Table 2 using another reference group. Instead of categorizing based on two factors (lipid level and lipid-lowering medication), I suggest the authors to stratify on use of lipid-lowering medication for each lipid parameter (exposure). In that way there will only be one exposure. Perhaps the message will be clearer then.

According to the American Diabetes Association, diabetic polyneuropathy is a diagnosis of exclusion (Pop-Busui et al. Diabetes Care 2017). Thus, before a diagnosis of neuropathy can be made, clinicians should exclude other causes for example chemotherapy, alcohol, hereditary, B-vitamin deficiency etc. It is unclear whether study participants were excluded if they had another cause of neuropathy than diabetes.

The authors state that their regression analyses were adjusted for all confounders defined as covariates that showed a significant correlation. Talking about confounders in a causal framework requires that confounders are separated from mediators and clearly described to the reader. If the aim of the paper is a casual interpretation, the researchers are interested in the “total effect” and would therefore avoid to adjust for mediators. Mediators may also associate “significantly” with the outcome and this rule can therefore not be used to interpret what covariates to adjust for. How did the authors ensure that they only adjusted for confounders and not mediators? Could there be residual confounding?

The cohort has a very high median age. Is it possible that those with high cholesterol/triglycerides have died before they had the possibility to participate in your study? Could this be a highly selective cohort and, in that way, cause an underestimation of the association?

Minor:

• It is unclear from the method section how many patients were eligible and how many patients were excluded. For example, patients were excluded due to missing lipid measurements, but it is unknown how many it was.

• It is unclear from the abstract what the main finding was. Please provide odds ratios.

• I would recommend not to show P-values. P-values and statistical testing should not be used to evaluate associations (Greenland et al. Eur J Epidemiology 2016). In addition, they should not be used to test differences for baseline characteristics.

• Abbreviations are not explained. Please write out the word the first time it is mentioned and include the abbreviation in parenthesis. Use the abbreviation thereafter. For example “triglycerides (TG)”.

• Please indicate with percent (%) how many in an exposure subgroup had neuropathy.

• The many abbreviations make the result section hard to read and understand. Try to rewording the subgroups.

• The word “risk” implies that the study looks at incident cases and can determine causality. Please use another word for example “association” or “odds”.

• Please use updated guidelines. The guideline on blood cholesterol has changed very much since 2013 (see discussion).

• The section about limitations lacks thoughts about residual confounding, selection bias, misclassification and reverse causality.

Reviewer #5: The research is thought of and appropriately carried out. The findings are consistent with the objectives of the study. The limitations reflect a good understanding of what the outcome should be.

7. PLOS authors have the option to publish the peer review history of their article (what does this mean?). If published, this will include your full peer review and any attached files.

Reviewer #2: No

Reviewer #3: No

Reviewer #4: No

Reviewer #5: **Yes: **Jimoh, Ahmed Kayode

---

## [Author Response · Author response to Decision Letter 1]

13 May 2023

Dear Editors and Reviewers, 

Thank you very much for your kind consideration of the publication of our article entitled “The association between hyperlipidemia, lipid-lowering drugs and diabetic peripheral neuropathy in patients with type 2 diabetes mellitus”. We deeply appreciate the editors’ and reviewers’ valuable comments and have revised our manuscript based on those opinions. For your convenience, the responses to the referees’ comments are shown in red color, and the changes are also shown in red color in the revised version of the manuscript. We would be glad to respond quickly to any further questions or comments.

Best regards,

Ming-Hong Chang, MD

 

Reviewer #2: 

The Authors of the manuscript entitled "The association between hyperlipidemia, lipid-lowering drugs and diabetic peripheral neuropathy in patients with type 2 diabetes mellitus" have made corrections according to the Reviewer's suggestions.

Ans. Thank you for your feedback. We appreciate your positive comments on our research.

Reviewer #3: 

Dear Authors and Editor,

The main problem of this manuscript is the cross-sectional analyses. Such design does not allow to evaluate any causal link between lipid profile or lipid lowering therapy and DPN. Indeed, this study is biased by several factors, e.g., confounding by indications, reverse causality, immortal time bias....

This design does not allow to address the role of lipid profile and LLT on DPN.

The authors should design a longitudinal study, evaluating onset of DPN after initiation of LLT or after exposure to low or high levels of HDL, TG, LDL etc...

The results therefore do not support the conclusions.

Ans. Thank you for your valuable comments. Our research aimed to investigate the association between hyperlipidemia exposure and lipid-lowering therapy with diabetic peripheral neuropathy among Chinese population. As a cross-sectional study, it is inherently difficult to confirm causal relationship. We have revised the manuscript to clarify that our study only suggests an association, but not causality. The revision is shown in red color at lines 426-427 in the revised manuscript. We appreciate your suggestion for a longitudinal study design and we are currently working on it.

Reviewer #4: 

Thank you for giving me the opportunity to read your paper. 

This is an interesting cross-sectional study of the association between hyperlipidemia, lipid-lowering drugs and diabetic peripheral neuropathy (DPN) in patients with type 2 diabetes mellitus (T2D). Despite studies conducted among Western cohorts have shown that lipid-lowering medication is not associated with DPN, the evidence is insufficient for Asian cohorts. The study included patients with type 2 diabetes seen in an outpatient clinic at a Taiwanese tertiary medical center. Upon enrollment baseline characteristics were collected including anthropometric measures, medical history, and medication. Neuropathy was evaluated using measurements recommend by clinical guidelines including answering the Michigan Neuropathy Screening Instrument and a thorough clinical examination. Using regression analyses, the authors report that neither hyperlipidemia nor lipid-lowering drugs were associated with a risk of DPN. Despite these findings, the study has limitations – please see my comments below.

Major comments:

1. For those in lipid-lowering drug treatment, the authors obtained the baseline lipid level within five years before the drug was prescribed. I understand that patients in lipid-lowering therapy used the medication at baseline. What was the purpose by defining the baseline lipid level as the level before drug treatment and not using the on-treatment lipid level? The on-treatment lipid level may be more informative since lipid-lowering therapy per definition impacts the lipid level. Furthermore, is it possible to obtain information on how longtime patients have received lipid-lowering therapy?

Ans. Thank you very much for your useful suggestion to make the manuscript more comprehensive. We defined the baseline lipid level as the level before lipid-lowering drug treatment because we hypothesized that hyperlipidemia contributes to microvasculopathies including DPN, and the process is likely to require prolonged exposure. Therefore, we considered the average lipid level in the five years prior to statin initiation as a more comprehensive representation of the patients' baseline lipid status, rather than using the lipid level at the time of statin initiation. 

Regarding the duration of lipid-lowering therapy, we could only obtain information on the patients' lipid-lowering medication use during the follow-up period, but not able to obtain how long time patients exactly received lipid-lowering therapy, limited by cross-sectional design. 

2. The cut-offs for triglycerides are very high. Other studies have used 150 mg/dL (1.7 mmol/L). 200 mg/dL is a target provided for clinicians who should consider initiating statin therapy in patients who have a high risk of CVD (Mach et al. 2019, 2019 ESC/EAS guidelines for the management of dyslipidaemias: Lipid modification to reduce cardiovascular risk). The cut-off is therefore not DPN specific. Furthermore, using cut-off points can be informative, but may also be problematic because the categorization implies an assumption about patients in a giving category have the same risk which may lead to loss of information (Royston et al. Stat med. 2006). While this assumption maybe approximately true in some cases, it is likely rare that this would hold biologically. Did the authors consider to use more flexible models to be inform the reader about the form of the association for example by using restricted cubic splines?

Ans. Thank you very much for your useful suggestion to make the manuscript more comprehensive. Due to the timeline of our study, which was conducted in 2013, we used the cut-off value of 200 mg/dL for hypertriglyceridemia based on the Taiwan National Health Insurance reimbursement regulations and the National Cholesterol Education Program Adult Treatment Panel III guidelines that were relevant at that time. This cut-off value was in line with clinical practice and treatment recommendations for hypertriglyceridemia in Taiwan during that period (lines 125-126 in the revised manuscript). Besides, it is worth mentioning that in our cohort, both baseline and follow-up serum triglyceride levels were comparable in patients with and without DPN, which suggest that triglyceride might not play an important role in our DPN cohort, regardless of which triglyceride threshold were applied.

We acknowledge that the use of cut-off points has limitations and may not fully capture the biological variability of the association. However, due to the constraints of our study design, we did not explore more flexible models such as restricted cubic splines. We will take this suggestion into consideration for future research.

3. It is highly problematic that the authors categorize patients into “hyperlipidemic” subgroups based on three very different lipid measurements. While LDL cholesterol is important as a cardiovascular disease risk marker, triglycerides may be the most important for neuropathy. When categorized according to EITHER cholesterol OR triglycerides the results will tend to be diluted because LDL cholesterol is not associated with neuropathy. The authors should therefore consider regrouping the patients and analyze the results for each lipid parameter separately. At last, it is unclear why the authors repeat the analysis in Table 2 using another reference group. Instead of categorizing based on two factors (lipid level and lipid-lowering medication), I suggest the authors to stratify on use of lipid-lowering medication for each lipid parameter (exposure). In that way there will only be one exposure. Perhaps the message will be clearer then.

Ans. Thank you very much for your useful suggestion to make the manuscript more comprehensive.

 1). Based on the literature review in the discussion section, in addition to triglycerides (TG), previous studies have also suggested associations between serum total cholesterol (TC) and low density lipoprotein cholesterol (LDL-C) with diabetic peripheral neuropathy (DPN), although the findings were not consistent (some studies reported positive associations, while others reported negative associations). Therefore, we decided to investigate not only the impact of TG but also TC and LDL-C. In the initial grouping, we defined hyperlipidemia as having either high TG, high LDL-C, or hyper TC. We understand your feedback that different types of lipid profiles may have different pathophysiological mechanisms influencing the complications of diabetes, so we conducted subgroup analyses for different lipid profiles separately in Table 3 to examine if they are risk factors for DPN. The reason why only TC and LDL-C were discussed in Table 3 is that TG did not show significant differences between the two groups of T2D patients with and without DPN in Table 1, so we conducted subgroup analyses for TC and LDL-C, which showed significant differences.

 2). In Table 2, the reason for using normolipidemic non-LLT users and normolipidemic LLT users as reference groups separately is that using normolipidemic non-LLT users as a reference group does not allow direct comparison of the DPN risk between normolipidemic LLT users and hyperlipidemic LLT users.

 3). We appreciate your suggestion for grouping, but using separate stratification for each lipid parameter based on medication use would result in only examining the association between each lipid parameter and DPN, without considering the combined effect of lipid status and medication use on DPN, which is what we intended to investigate.

4. According to the American Diabetes Association, diabetic polyneuropathy is a diagnosis of exclusion (Pop-Busui et al. Diabetes Care 2017). Thus, before a diagnosis of neuropathy can be made, clinicians should exclude other causes for example chemotherapy, alcohol, hereditary, B-vitamin deficiency etc. It is unclear whether study participants were excluded if they had another cause of neuropathy than diabetes.

Ans. Thank you very much for your useful suggestion to make the manuscript more comprehensive. Before enrollment in the study, the physicians and trained care-management nurse had screened all the potential participants and excluded those who had a history of chemotherapy exposure, alcohol abuse, or hereditary neuropathy. We have revised our manuscript to include this information. The revision is shown in red color at lines 97-99 in the revised manuscript.

We did not perform routine laboratory tests for vitamin B deficiency or other exposure to toxins in our cohort. However, we reviewed many other well-known studies that we have cited (e.g., Nielsen and Nordestgaard, 2014[1]; Van Acker et al., 2009[2]), and many of them did not exclude these conditions. Thank you for your suggestion and we will consider it in future studies.

5. The authors state that their regression analyses were adjusted for all confounders defined as covariates that showed a significant correlation. Talking about confounders in a causal framework requires that confounders are separated from mediators and clearly described to the reader. If the aim of the paper is a casual interpretation, the researchers are interested in the “total effect” and would therefore avoid to adjust for mediators. Mediators may also associate “significantly” with the outcome and this rule can therefore not be used to interpret what covariates to adjust for. How did the authors ensure that they only adjusted for confounders and not mediators? Could there be residual confounding?

Ans. Thank you very much for your useful suggestion to make the manuscript more comprehensive. We agree that it is crucial to clearly distinguish between confounders and mediators in a causal framework. To address the confounders, we followed the criteria for defining confounders as outlined in a previous publication by Michael A Babyak (2009)[3], confounders need to meet three criteria: (1) being a risk factor of the outcome (in our study, DPN), (2) being associated with the exposure (in our study, hyperlipidemia and LLT), and (3) should not be part of the causal pathway from exposure to outcome. Regarding mediators, there are also three criteria to consider. The first two are the same as those for confounders, but the third is that the mediator must be a presumed causal consequence of the exposure, meaning that the exposure causes the mediator, which in turn causes the outcome. 

To ensure that we only adjusted for confounders and not mediators, we followed these steps:

 1). We used Fisher’s exact test, chi-squared test, and ANOVA tests to compare the variables in the study population with DPN (outcome) and identified variables that were associated with DPN. These variables were considered as the potential confounders in our study.

 2). Based on criterion 1 for confounders, we reviewed previous researches and ensured that the following potential confounders, such as age[4, 5], waist[5], smoking[4], duration of diabetes[4, 6], type of diabetes treatment[6], eGFR[7], and heart disease[8], were reported as risk factors for DPN. Besides, because gender is an important demographic variable and was also known as an independent risk factor for DPN[9], we decided to add gender in the model of multivariate regression analysis.

 3). We confirmed that these variables were not part of the causal pathway from exposure to outcome. Through our understanding of the disease pathophysiology, we excluded the possibility that exposure (hyperlipidemia & LLT) caused these variables, which in turn caused DPN.

We acknowledge that residual confounding may still exist despite our efforts to identify and adjust for confounders. However, we believe that our approach to adjusting for confounders was appropriate based on current best practices in causal inference.

6. The cohort has a very high median age. Is it possible that those with high cholesterol/triglycerides have died before they had the possibility to participate in your study? Could this be a highly selective cohort and, in that way, cause an underestimation of the association?

Ans. Thank you very much for your useful suggestion to make the manuscript more comprehensive. Our cohort was not highly selected, and the inclusion criteria were all adults aged 20 years or older diagnosed with type 2 diabetes in 2013, with exclusion of patients with type 1 diabetes, gestational diabetes, chemotherapy exposure, alcohol abuse or hereditary neuropathy. According to studies based on the Taiwan's National Health Insurance Research Database (Sheen YJ, et al. Journal of the Formosan Medical Association. 2019)[10], the prevalence of type 2 diabetes among those aged 20-79 years increased by 41% between 2005 and 2014, and more than half of the type 2 diabetes patients were over the age of 65 in 2014. Therefore, the relatively high median age in our cohort may be attributed to the aging of the diabetic population in Taiwan.

Minor comments:

1. It is unclear from the method section how many patients were eligible and how many patients were excluded. For example, patients were excluded due to missing lipid measurements, but it is unknown how many it was.

Ans. Thank you very much for your useful suggestion to make the manuscript more comprehensive. We have updated the Result section and included a flow chart for patient selection. This flow chart provides an overview of the number of patients initially eligible, the reasons for exclusion, and the final cohort size used in our analysis. It shows that out of the 2,838 patients screened, 85 were excluded due to having type 1 diabetes or gestational diabetes, and 305 were excluded due to missing serum lipid measurements. The final cohort included 2,448 patients for analysis.

2. It is unclear from the abstract what the main finding was. Please provide odds ratios.

Ans. Thank you very much for your useful suggestion to make the manuscript more comprehensive. Due to the detailed and multifaceted comparisons we conducted in our study, including analyses of hyperlipidemia status, lipid-lowering therapy (LLT) use, and subgroup analyses of various lipid parameters and types of LLT, we found no significant differences in the risk of diabetic peripheral neuropathy (DPN) across these groups. Due to limited word count, the odd ratios were not provided initially. The revision is shown in red color at lines 33-39 in the revised manuscript.

3. I would recommend not to show P-values. P-values and statistical testing should not be used to evaluate associations (Greenland et al. Eur J Epidemiology 2016). In addition, they should not be used to test differences for baseline characteristics.

Ans. Thank you very much for your useful suggestion to make the manuscript more comprehensive. We appreciate your advice on the limitations of using P-values to evaluate associations and test differences for baseline characteristics, as referenced in the study by Greenland et al. in Eur J Epidemiology (2016)[11]. Based on your suggestion, we have made adjustments in our manuscript and have refrained from displaying the actual numerical values of P-values. Instead, we have indicated significant findings with an asterisk (*) for characteristics with P-values < 0.05. We believe that this approach addresses your concerns while still providing relevant information in our results. Thank you for bringing this to our attention.

4. Abbreviations are not explained. Please write out the word the first time it is mentioned and include the abbreviation in parenthesis. Use the abbreviation thereafter. For example, “triglycerides (TG)”.

Ans. Thank you very much for your useful suggestion to make the manuscript more comprehensive. We have reviewed the manuscript again to ensure that all abbreviations are explained in full the first time they are mentioned, with the abbreviation included in parentheses. We will also ensure that the abbreviations are used consistently throughout the manuscript thereafter.

5. Please indicate with percent (%) how many in an exposure subgroup had neuropathy.

Ans. Thank you very much for your useful suggestion to make the manuscript more comprehensive. We have provided detailed information on the number of patients with DPN and their percentages in the Results section for each exposure subgroup, including normolipidemic non-LLT users, normolipidemic LLT users, and hyperlipidemic LLT users. 

The updated paragraph now reads as follows: "The normolipidemic non-LLT user group comprised 396 patients (16.2%), 98 of whom (24.75%) had DPN; the normolipidemic LLT user group comprised 946 patients (38.6%), 213 of whom (22.52%) had DPN; and the hyperlipidemic LLT user group comprised 1106 individuals (45.2%), 213 of whom (19.26%) had DPN." The revision is shown in red color at lines 230-234 in the revised manuscript.

6. The many abbreviations make the result section hard to read and understand. Try to rewording the subgroups.

Ans. Thank you very much for your useful suggestion to make the manuscript more comprehensive. We have addressed the issue of excessive abbreviations of the manuscript. The modified abbreviations used to describe the subgroups are as follows:

• Normolipidemic non-LLT users: Referring to patients with normal blood lipid levels who did not receive lipid-lowering therapy (LLT).

• Normolipidemic LLT users: Referring to patients with normal serum lipid levels who received LLT.

• Hyperlipidemic LLT users: Referring to patients with elevated serum lipid levels who received LLT.

7. The word “risk” implies that the study looks at incident cases and can determine causality. Please use another word for example “association” or “odds”.

Ans. Thank you very much for your useful suggestion to make the manuscript more comprehensive. We have revised the wording in the manuscript to avoid using the term "risk" and have replaced it with "association" or "odds" to better reflect the study design and minimize confusion.

8. Please use updated guidelines. The guideline on blood cholesterol has changed very much since 2013 (see discussion).

Ans. Thank you very much for your useful suggestion to make the manuscript more comprehensive. We acknowledge that there have been significant updates in cholesterol management guidelines since 2013. However, in the Discussion section of our manuscript, we mentioned that “Based on the guideline of blood cholesterol management from American College of Cardiology/American Heart Association (ACC/AHA) in 2013, statin is recommended in diabetic patients whose serum LDL levels are between 70 to 189 mg/dL.” This was included to provide context for the time period of our study, which enrolled patients in 2013, and to explain why many patients without hyperlipidemia received LLT in our cohort. We appreciate your feedback and will consider using the most updated guidelines for any future follow-up studies.

9. The section about limitations lacks thoughts about residual confounding, selection bias, misclassification and reverse causality.

Ans. Thank you very much for your useful suggestion to make the manuscript more comprehensive. In the limitations section of our manuscript, we did not specifically mention residual confounding, selection bias, misclassification, and reverse causality, which are common limitations in observational studies. However, we employed standardized data collection procedures and gathered detailed information on potential confounding factors, laboratory test values, and diabetes treatment to minimize the presence of residual confounding. Additionally, we used a validated clinical screening instrument (MNSI) to reduce the likelihood of misclassification. We acknowledged that these limitations are important considerations in our study. We revised our manuscript to reflect our limitations in current study. The revision is shown in red color at lines 427-434 in the revised manuscript.

 

Reviewer #5: 

The research is thought of and appropriately carried out. The findings are consistent with the objectives of the study. The limitations reflect a good understanding of what the outcome should be.

Ans. Thank you for your feedback. We appreciate your positive comments on our research.

 

Reference

1. Nielsen SF, Nordestgaard BG. Statin use before diabetes diagnosis and risk of microvascular disease: a nationwide nested matched study. Lancet Diabetes Endocrinol. 2014;2(11):894-900. Epub 20140909. doi: 10.1016/s2213-8587(14)70173-1. PubMed PMID: 25217178.

2. Van Acker K, Bouhassira D, De Bacquer D, Weiss S, Matthys K, Raemen H, et al. Prevalence and impact on quality of life of peripheral neuropathy with or without neuropathic pain in type 1 and type 2 diabetic patients attending hospital outpatients clinics. Diabetes Metab. 2009;35(3):206-13. Epub 20090317. doi: 10.1016/j.diabet.2008.11.004. PubMed PMID: 19297223.

3. Babyak MA. Understanding confounding and mediation. Evid Based Ment Health. 2009;12(3):68-71. doi: 10.1136/ebmh.12.3.68. PubMed PMID: 19633239.

4. Liu X, Xu Y, An M, Zeng Q. The risk factors for diabetic peripheral neuropathy: A meta-analysis. PLoS One. 2019;14(2):e0212574. Epub 20190220. doi: 10.1371/journal.pone.0212574. PubMed PMID: 30785930; PubMed Central PMCID: PMCPMC6382168.

5. Ziegler D, Rathmann W, Dickhaus T, Meisinger C, Mielck A. Prevalence of polyneuropathy in pre-diabetes and diabetes is associated with abdominal obesity and macroangiopathy: the MONICA/KORA Augsburg Surveys S2 and S3. Diabetes Care. 2008;31(3):464-9. Epub 20071126. doi: 10.2337/dc07-1796. PubMed PMID: 18039804.

6. Mørkrid K, Ali L, Hussain A. Risk factors and prevalence of diabetic peripheral neuropathy: A study of type 2 diabetic outpatients in Bangladesh. Int J Diabetes Dev Ctries. 2010;30(1):11-7. doi: 10.4103/0973-3930.60004. PubMed PMID: 20431800; PubMed Central PMCID: PMCPMC2859278.

7. Wang CS, Pai YW, Lin CH, Lee IT, Chang MH. Renal impairment is one of appropriate predictors of future diabetic peripheral neuropathy: a hospital-based 6-year follow-up study. Sci Rep. 2022;12(1):5240. Epub 20220328. doi: 10.1038/s41598-022-09333-3. PubMed PMID: 35347201; PubMed Central PMCID: PMCPMC8960825.

8. Fakkel TM, Çakici N, Coert JH, Verhagen AP, Bramer WM, van Neck JW. Risk Factors for Developing Diabetic Peripheral Neuropathy: a Meta-analysis. SN Comprehensive Clinical Medicine. 2020;2(10):1853-64. doi: 10.1007/s42399-020-00480-0.

9. Group TS. Risk Factors for Diabetic Peripheral Neuropathy in Adolescents and Young Adults With Type 2 Diabetes: Results From the TODAY Study. Diabetes Care. 2021;45(5):1065-72. Epub 20211029. doi: 10.2337/dc21-1074. PubMed PMID: 34716210; PubMed Central PMCID: PMCPMC9174958.

10. Sheen YJ, Hsu CC, Jiang YD, Huang CN, Liu JS, Sheu WH. Trends in prevalence and incidence of diabetes mellitus from 2005 to 2014 in Taiwan. J Formos Med Assoc. 2019;118 Suppl 2:S66-s73. Epub 20190709. doi: 10.1016/j.jfma.2019.06.016. PubMed PMID: 31300322.

11. Greenland S, Senn SJ, Rothman KJ, Carlin JB, Poole C, Goodman SN, et al. Statistical tests, P values, confidence intervals, and power: a guide to misinterpretations. Eur J Epidemiol. 2016;31(4):337-50. Epub 20160521. doi: 10.1007/s10654-016-0149-3. PubMed PMID: 27209009; PubMed Central PMCID: PMCPMC4877414.

---

## [Decision Letter · Decision Letter 2]

5 Jun 2023

The association between hyperlipidemia, lipid-lowering drugs and diabetic peripheral neuropathy in patients with type 2 diabetes mellitus

PONE-D-22-23997R2

Dear Dr. Chang,

We’re pleased to inform you that your manuscript has been judged scientifically suitable for publication and will be formally accepted for publication once it meets all outstanding technical requirements.

Kind regards,

Fredirick Lazaro mashili, MD, PhD

Academic Editor

PLOS ONE

Additional Editor Comments (optional):

Generally, the authors have sufficiently responded to all the comments given by reviewers. Now, the manuscript looks generally well-organized and clearly presents the study's methodology, results, and implications. The Discussion section effectively contextualizes the findings, comparing them with existing research and thoroughly discussing potential explanations for the observed results. Even though the limitations section could have been further elaborated to address the potential impact of these limitations on the study's findings and conclusions, the authors have tried to address the limitations of their findings.

Overall, the study contributes valuable insights to the understanding of DPN in T2D Asian patients, emphasizing that DPN is a multifactorial condition and suggesting that controlling hyperlipidemia alone may not be sufficient to prevent DPN.

Reviewers' comments:

Reviewer's Responses to Questions

**Comments to the Author**

1. If the authors have adequately addressed your comments raised in a previous round of review and you feel that this manuscript is now acceptable for publication, you may indicate that here to bypass the “Comments to the Author” section, enter your conflict of interest statement in the “Confidential to Editor” section, and submit your "Accept" recommendation.

Reviewer #2: All comments have been addressed

Reviewer #3: (No Response)

Reviewer #4: All comments have been addressed

2. Is the manuscript technically sound, and do the data support the conclusions?

Reviewer #2: Yes

Reviewer #3: No

Reviewer #4: Yes

3. Has the statistical analysis been performed appropriately and rigorously? 

Reviewer #2: Yes

Reviewer #3: No

Reviewer #4: I Don't Know

4. Have the authors made all data underlying the findings in their manuscript fully available?

Reviewer #2: Yes

Reviewer #3: Yes

Reviewer #4: No

5. Is the manuscript presented in an intelligible fashion and written in standard English?

Reviewer #2: Yes

Reviewer #3: Yes

Reviewer #4: Yes

6. Review Comments to the Author

Reviewer #2: # Review

The Authors have made corrections according to the Reviewer's suggestions and in the present form the manuscript may be suitable for the publication.

Reviewer #3: As said before the cross sectional design is not adequate for the purposes of this study. The relationship between onset of complications and initiation of treatments is not considered, multiple bias can influence the reported association.

Reviewer #4: (No Response)

7. PLOS authors have the option to publish the peer review history of their article (what does this mean?). If published, this will include your full peer review and any attached files.

Reviewer #2: No

Reviewer #3: No

Reviewer #4: No

---

## [Editor Report · Acceptance letter]

7 Jun 2023

PONE-D-22-23997R2 

The association between hyperlipidemia, lipid-lowering drugs and diabetic peripheral neuropathy in patients with type 2 diabetes mellitus 

Dear Dr. Chang:

I'm pleased to inform you that your manuscript has been deemed suitable for publication in PLOS ONE. Congratulations! Your manuscript is now with our production department. 

Kind regards, 

on behalf of

Dr Fredirick Lazaro mashili 

Academic Editor

PLOS ONE